# Potential drivers of the recent large Antarctic ozone holes

Hannah E. Kessenich [1], Annika Seppälä [1] ✉ & Craig J. Rodger[1]

The past three years (2020–2022) have witnessed the re-emergence of large, long-lived ozone holes over Antarctica. Understanding ozone variability remains of high importance due to the major role Antarctic stratospheric ozone plays in climate variability across the Southern Hemisphere. Climate change has already incited new sources of ozone depletion, and the atmospheric abundance of several chlorofluorocarbons has recently been on the rise. In this work, we take a comprehensive look at the monthly and daily ozone changes at different altitudes and latitudes within the Antarctic ozone hole. Following indications of early-spring recovery, the October middle stratosphere is dominated by continued, significant ozone reduction since 2004, amounting to 26% loss in the core of the ozone hole. We link the declines in mid-spring Antarctic ozone to dynamical changes in mesospheric descent within the polar vortex, highlighting the importance of continued monitoring of the state of the ozone layer.

The Montreal Protocol is widely considered to be successful[1], and observed levels of controlled ozone depleting substances (ODSs) in the stratosphere, quantified as equivalent effective stratospheric chlorine (EESC), have been slowly declining since the late 1990's[1–3]. Recovery of the Antarctic ozone hole in response to declining EESC is of particular interest, as the hole's dramatic development in the late 20th century has been directly linked to anthropogenic ODS emissions[4–6]. For as long as large ozone holes persist, climate variability across the Southern Hemisphere (SH) will continue to be impacted[7–9].

Part of assessing the recovery of the Antarctic ozone hole requires acknowledgement of other factors besides EESC which contribute to anomalously high or low ozone values. Springtime temperature and wind patterns greatly impact SH ozone hole development, along with aerosol loading from wildfires and volcanic eruptions, as well as changes in the solar cycle[1,10–21]. Trends in greenhouse gas emissions are also expected to impact ozone recovery[22–24], and proposed climate change mitigation strategies such as stratospheric aerosol injection would further impact recovery dates, if implemented[1]. Taking ozone modulating factors into account, the 2022 Scientific Assessment of Ozone Depletion[1] concluded that the Antarctic ozone hole should be on track to recover by 2065. Latest results[25,26], however, indicate that recovery may be delayed due to previously unaccounted for chlorine release from wildfire aerosols and anthropogenic emissions.

Nevertheless, the 2022 Antarctic spring season saw yet another large ozone hole with an extent and duration remarkably similar to the large holes of 2020 and 2021[1,27]. Considering that 2015 and 2018 were also similar record years (five of the past eight years overall have exhibited record ozone holes), our aim here is to better understand recent changes in Antarctic ozone.

In this work, we take a systematic approach to analyse the behaviour of SH polar ozone throughout the key springtime months of September to November. We begin by examining the recent record ozone hole years of 2020–2022 in the context of historical total column ozone (TCO) observations and onset/breakup metrics. We then focus on signs of ozone recovery in vertical and horizontal distributions of SH polar stratospheric ozone during the SH spring months. Finally, we investigate links between the ozone signals and dynamical changes in order to identify the likely drivers of the observed ozone changes.

## Results
### Total column ozone and onset/breakup dates
The Antarctic ozone hole emerges in the early austral spring, during the month of August[1]. It typically persists until the end of November, coinciding with the breakdown of the SH polar vortex[1]. Year-to-year trends in September SH ozone have been considered to be the most

[1]Department of Physics, University of Otago, Dunedin, New Zealand. ✉e-mail: annika.seppala@otago.ac.nz

sensitive for assessing ozone recovery directly due to the reduction of EESC, as this month is largely dominated by chemical, rather than dynamical, variability[1,12,28–32]. While October and November are thought to be more influenced by dynamics, they are important for assessing the full extent of the ozone hole area and depth[12,28,33] as well as its dynamical impacts[34].

Considering the large Antarctic ozone holes of 2020–2022[27], we assess the overall changes in September, October, and November SH TCO from 2001 to 2022. Figure 1a presents the TOMS/OMI polar mean TCO (see Methods), with the years from 1979 included for context. The analysis is analogous to that previously reported for November by Zambri et al.[34]. We use a simple linear regression rather than a multivariate fit, as we focus on long term changes in ozone rather than attribution. Henceforth, we refer to the slope of the linear regression fit as the long term "change" in ozone instead of "trend", analogous to Orbe et al.[35].

Figure 1a highlights that the monthly mean TCO values have little year-to-year variability in 2020–2022. While the September TCO values are low in the most recent years, overall, September continues to display a slightly upwards slope (red line in Fig. 1a). This slope generally aligns with the expected recovery behaviour. However, this is not the case for October and November (blue and cyan lines in Fig. 1a), which show a negative change in TCO since 2001. Importantly for all months, the slope of the linear fit is statistically insignificant. As in previous studies[34], we also find that the choice of turnaround/pivot year for the overall ozone trends (e.g., 2000 vs 2001) has little impact on the trajectory. This raises the question of how many years of observational records will be needed to solidify TCO recovery, particularly in light of emerging new sources of chlorine and the changing state of the Antarctic stratosphere due to climate change influences.

Additionally, we assess the ozone hole onset and breakup dates from 2005–2022, using a TCO depth threshold of < 130 Dobson Units (DU) and an area threshold of 1 million km² [28] (see Methods). We choose the 130 DU threshold (as opposed to 175/220 DU) in order to focus on the deepest levels of ozone depletion, characteristic of recent years[27]. Similar to Stone et al.[28], we identify the ozone hole onset date

as the day when the ozone hole area exceeds the threshold for 3 days in a row. In addition to the onset date, we further identify the ozone hole breakup date as the day when the < 130 DU ozone hole no longer surpasses the 1 million km² area threshold. The ozone hole onset and breakup dates are presented in Fig. 1b. The trend in the onset date (green line in Fig. 1b) agrees with previous findings of a delayed ozone hole opening date[1,12,27,28,33]. This delay is likely an indication of early-spring recovery due to EESC reduction[1,12]. Our results for the breakup date trend (magenta line in Fig. 1b) show a similar delay in the breakup of the ozone hole, albeit much noisier and insignificant. Given that the deep ozone hole breakup typically occurs during October/November, a later breakup date is in agreement with the declines in October/November TCO in Fig. 1a. These results suggest that there may be new patterns emerging in the mid-spring evolution of the ozone hole, thus delaying the breakup date.

## Long term change in vertical ozone distribution

Previous studies have found evidence of vertically and latitudinally resolved patterns in mid-latitude ozone trends that can oppose each other at different stratospheric layers and, as a result, can be indistinguishable in column measurements[15,36–39]. Figure 2 shows the vertically and latitudinally resolved SH ozone change [ppmv/year] from 2004 to 2022 using MLS/Aura observations (see Methods). The change is based on a linear regression with time as the sole predictor and is determined individually for September-November, as before. Areas with positive/negative concentration changes indicate atmospheric layers which have an increased/decreased amount of ozone at their given latitude through the 19 years of MLS observations. We find consistent, significant positive (red, hatched) ozone change of up to 0.03 ppmv/year in the upper stratosphere (between about 1–7 hPa, or 34–46 km) throughout the spring months. This is focused equatorward of 60°S and is consistent with previously reported increases in ozone near the edge of the polar vortex[32]. Our results also show a region of significant reduction in ozone (blue, hatched), with a change of up to −0.09 ppmv/year, in October. This takes place at lower altitudes, in the middle stratosphere between 7 and 60 hPa (~20–34 km).

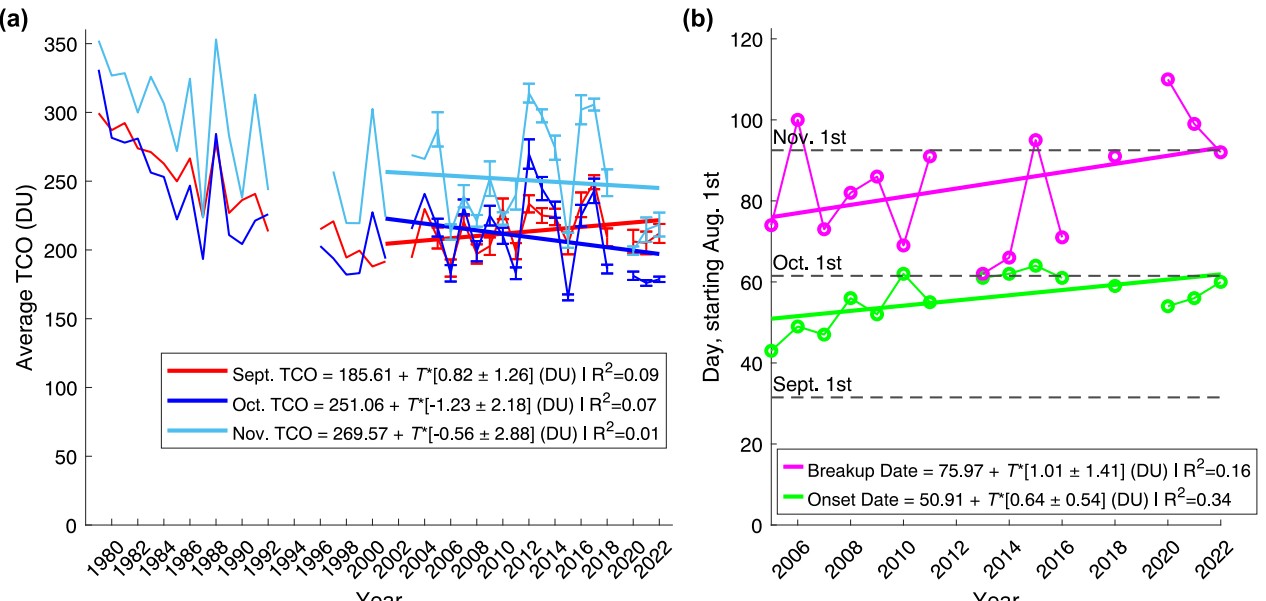

**Fig. 1 | Total column ozone (TCO) and ozone hole onset and breakup dates from TOMS/OMI observations. a** Zonal mean TCO in Dobson units (DU) across latitudes 60°S to 90°S shown for the months of Sept., Oct., and Nov. for years 1979–2022 (2002 and 2019 excluded). Error bars for OMI represent the standard deviation of the daily mean TCO measurements (2005–2022). The lines present a linear fit from 2001–2022, with the fit equation shown in the legend. The linear fit uncertainty is quoted at the 95% confidence interval and T = Years since 2001. **b** Deep ozone hole onset and breakup dates (130 DU, 1 million km²) for years 2005–2022. Missing years do not surpass the TCO/area threshold during the season. The lines present a linear fit from 2005–2022, with the fit equation shown in the legend. The linear fit uncertainty is quoted at the 95% confidence interval and T = Years since 2005.

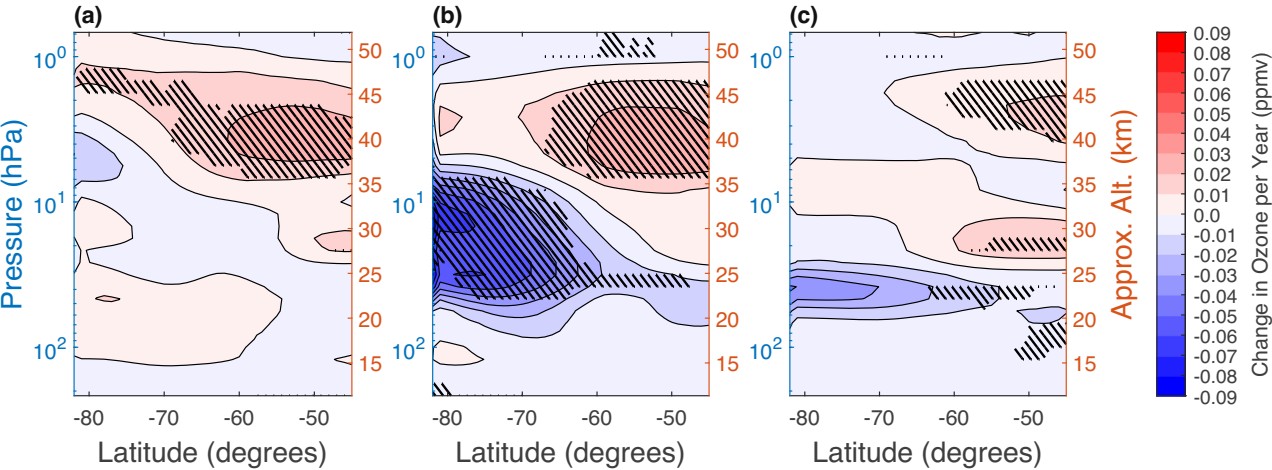

**Fig. 2 | Change in zonal mean MLS/Aura ozone observations for the period of 2004–2022.** Results are presented across latitudes 45°S–82°S and pressure levels 0.68–215 hPa (approximate altitudes in km given on the right) for the months of **a** September, **b** October, and **c** November. The coloured contours (ppmv/year, contour level intervals are 0.01 ppmv/year) represent the slope of a linear fit across the range of years. All volume mixing ratio changes that are significantly different from zero (at ≥95% level) are indicated with black hatching.

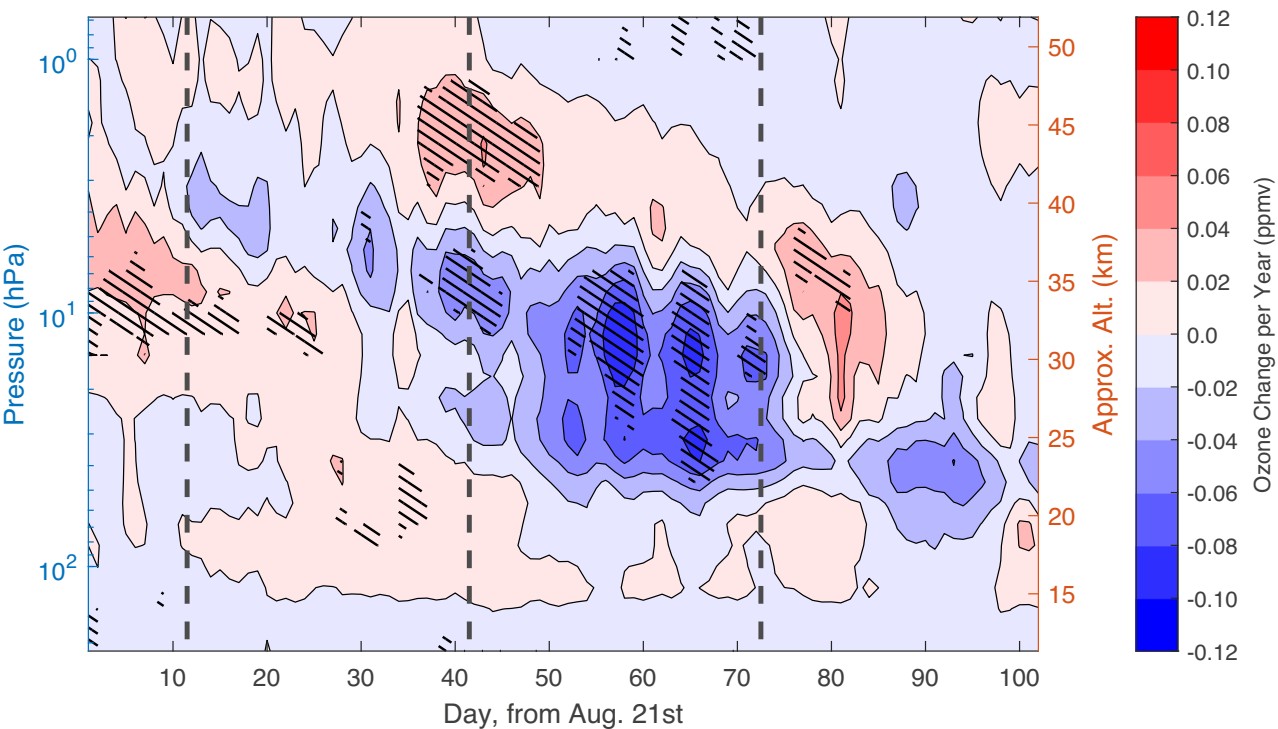

**Fig. 3 | Change in daily, zonally averaged MLS/Aura polar (75°S–82°S) ozone concentration (VMR), from Aug. 21st to Nov. 30th.** Black vertical lines indicate separation between months. A simple linear fit across the range of years (2004–2022) is found, with contour colours representing the slopes (ppmv/year). Changes that are significantly different from zero (at ≥95% level) are indicated with black hatching.

Unlike the recovery signal, the ozone reduction is focused on polar latitudes, poleward of 60°S.

To further determine the time evolution of the positive and negative changes found in the monthly analysis, we investigate day-to-day progression of the high latitude polar ozone volume mixing ratios (see Methods), with results shown in Fig. 3. We find a primarily insignificant positive signal (red) starting above 10 hPa in late August and extending to around 100 hPa by the end of September. This finding of a positive change in middle stratospheric ozone during late August and early September is in agreement with the results presented in Fig. 1, which show an insignificantly increasing September TCO trend and a

shift towards a later ozone hole onset date. A region of positive change (up to 0.04 ppmv/year) is also present in the upper stratosphere in September. This region extends into the first ten days October, after which it appears to descend to 3–26 hPa (~25–40 km) by the start of November. For the positive signals overall, the peak values are between 0.04 and 0.06 ppmv/year.

Contrary to the positive signals, the large area of negative (blue) change found during October (Fig. 2) is clearly present, and statistically significant, throughout the month of October. Most of the ozone loss signal in October is focused between 5 and 50 hPa (~21–37 km) and is linked to a feature that first appears in late August at around

1–2 hPa (~45 km). The regions of the highest negative ozone change reach upwards of −0.10 ppmv/year and are found to be statistically significant. Critically, this negative region overlaps with the main ozone layer, with highest ozone mixing ratios and therefore the core of the SH ozone layer, located between 4 and 20 hPa (~27–38 km)[40]. The finding of a negative change in middle stratospheric ozone during October is in agreement with the results presented in Fig. 1, which indicated decreasing October TCO and a shift towards a later breakup date of the deep ozone hole.

### Ozone change in the upper vs middle stratosphere

The findings in Figs. 2 and 3 support the notion that analyses conducted purely with TCO measurements obscure key information regarding vertical changes within the SH springtime ozone layer. We demonstrate this further by using MLS/Aura observations to construct a partial column ozone (PCO) time series over the regions of interest found previously. Figure 4 presents the October PCO timeseries for four distinct regions: Middle stratosphere (5–50 hPa, ~20–37 km) (1) polar latitudes (60°S–82°S), and (2) high polar latitudes (75°S–82°S), as well as upper stratosphere (1–5 hPa, ~37–50 km) (3) polar latitudes (60°S–82°S), and (4) near the vortex edge region (45°S–60°S).

Both PCO time series in the middle stratosphere show a significant negative trajectory. Overall, the net change of the time period from 2004 to 2022 is approximately −26% in the area closest to the pole (magenta line, 75°S–82°S). The PCO values for the 5–50 hPa region are generally between 80 and 160 DU, indicating that this region is a large contributor by weight to the corresponding TCO measurements. The values from higher altitudes, 1–5 hPa, tell a different story. This linear fit is slightly positive and only significant when the region of interest is limited to 45°S–60°S (light blue line). While the slope is positive, this increase is equivalent to less than 2 DU, which does not represent a notable contribution to the change in the total ozone column.

### Drivers of the changes in ozone

We now turn towards identifying potential drivers for the decline in October polar ozone in the 5–50 hPa vertical range, as identified in Figs. 2–4. While ozone depletion due to ODSs is known to begin in late August[1], October marks the month with the largest ozone hole extent and depth[1,12,28,33]. The month of October is also when dynamical variability is thought to begin to play a larger role in the ozone hole progression, particularly in the form of mesospheric air transport into the polar vortex[1,17,41,42].

We first look for a change in mesospheric descent inside the polar vortex by using carbon monoxide (CO) observations from MLS/Aura as an atmospheric tracer[43]. Similar to Fig. 3 for ozone, we perform a linear regression on the day-to-day progression of polar CO across the years 2004–2022. The results are presented in Fig. 5a. We find a tongue of increasing CO which reaches the same altitude (7–20 hPa, ~27–34 km) as the upper portion of the negative ozone change detected during the month of October (Fig. 3). This tongue sits above a region of negative change in CO (20–50 hPa, ~21–27 km), implying an upwards shift in altitude of the arriving mesospheric air mass.

To investigate the potential implications of this shift on the stratospheric chemical balance, we look at the relationship between CO and stratospheric nitrogen dioxide ($NO_2$) column (from OMI/Aura). $NO_2$ is a member of the $NO_x$ family of chemicals which are known to be transported into the polar vortex from the mesosphere, alongside CO, where $NO_x$ can in turn deplete stratospheric ozone[43,44]. Descending $NO_x$ is also known to curb chlorine-driven ozone loss through the Energetic Particle Precipitation (EPP) indirect effect when the $NO_x$ reaches the middle and lower stratosphere[1,17,42]. Figure 5b presents the correlation between the altitude of late-October CO descent and the average October polar stratospheric $NO_2$ (see Methods). We find a significant negative correlation ($R^2$ value of 0.78) between the CO altitude and stratospheric $NO_2$, with recent years (2020–2022)

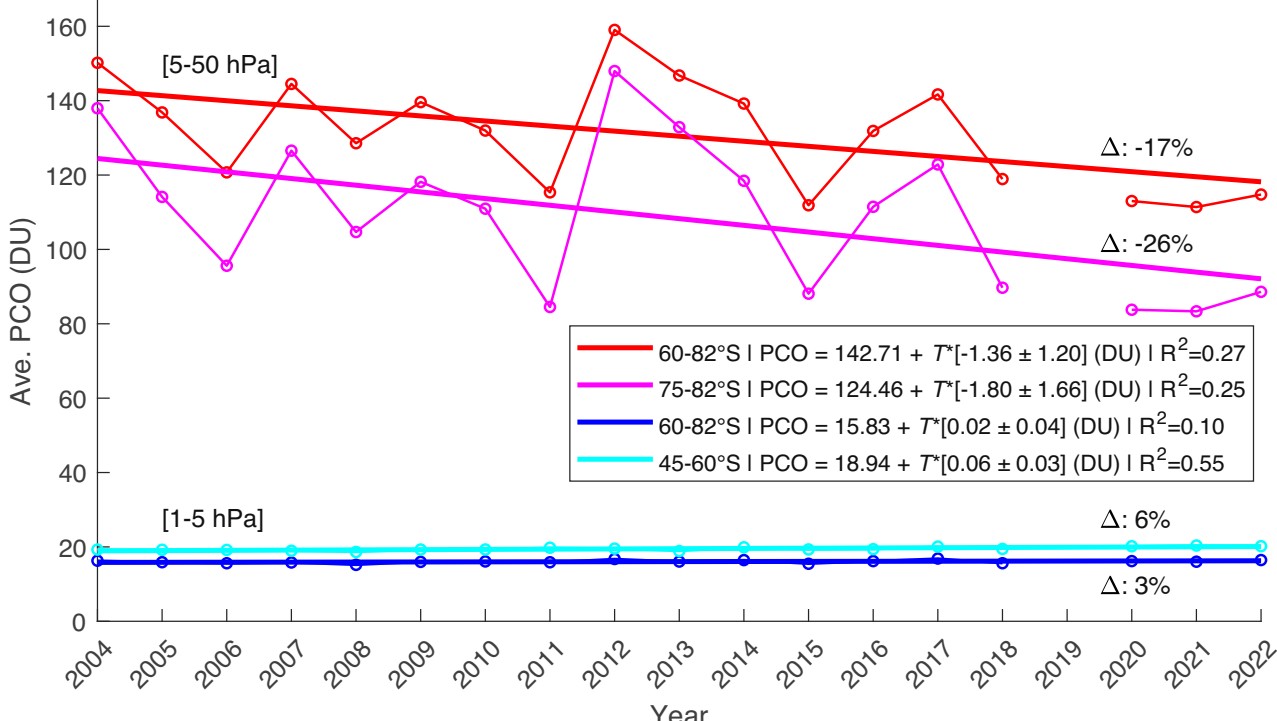

**Fig. 4 | October partial column ozone (PCO) over four altitude and latitude ranges of interest derived from MLS observations for 2004–2022.** Red line: PCO for the pressure levels of 5–50 hPa and averaged from 60°S to 82°S; Pink line: As red line, but now averaged from 75°S to 82°S; Dark blue line: PCO calculated for pressure levels 1–5 hPa and averaged from 60°S to 82°S; Light blue line: As dark blue line, but now averaged from 45°S to 60°S. The results from linear fits are shown in the legend, with errors quoted at 95% confidence interval. Here, $T$ = years since 2004, and the percent change in PCO since 2004 is given after the $\Delta$-symbol.

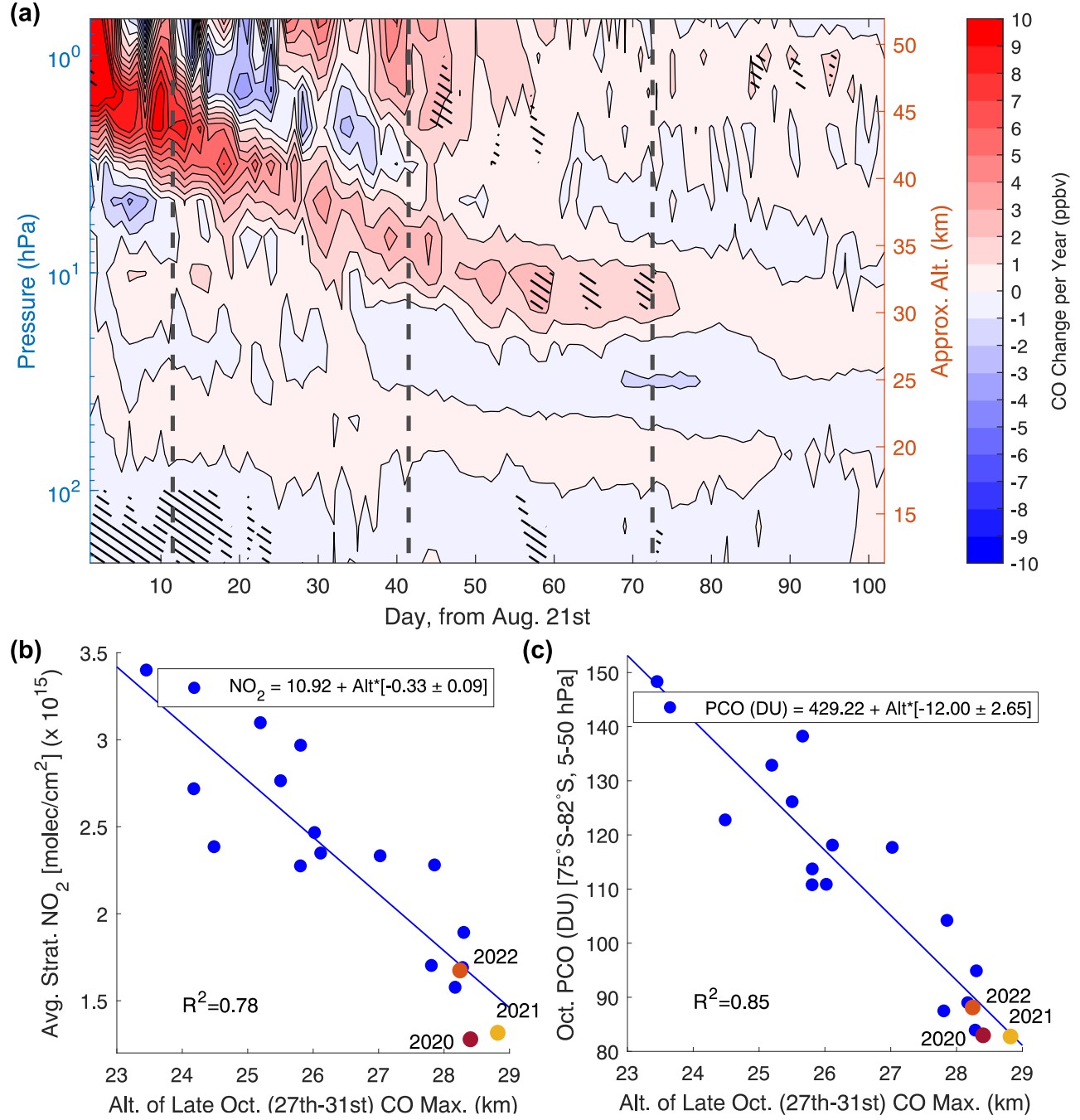

**Fig. 5 | Indicators of links between ozone and changes in carbon monoxide (CO) and nitrogen dioxide (NO₂).** **a** Change in daily, zonally averaged MLS/Aura polar (75°S–82°S) CO concentration (VMR), from Aug. 21st to Nov. 30th. Black vertical lines indicate separation between months. A simple linear fit across the range of years (2004–2022) is found, with contour colours representing the slopes (ppbv/ year). Changes that are significantly different from zero (at ≥95% level) are indicated with black hatching. **b** October mean stratospheric NO₂ vs. altitude of late-October CO maximum for 2005–2022 and **c** October partial column ozone (PCO) (from 5 to 50 hPa and 60°S–82°S) vs. altitude of late-October CO maximum for 2004–2022. The results from a linear fit for **b**, **c** are shown in the legends, with error quoted at 95% confidence interval and Alt = altitude of late Oct. CO maximum.

consistently corresponding to higher altitudes of CO and lower NO₂ column.

Finally, we correlate the altitude of late-October CO descent with the second PCO series from Fig. 4 (magenta line, 75°S–82°S, 5–50 hPa), with results presented in Fig. 5c. This too produces a significant negative correlation, with an R² value of 0.85. These results imply that the EPP indirect effect (or lack of arriving NO$_x$) due to changes in the timing and altitude of mesospheric descent could play a role in PCO

during October. Further investigation is required to determine the exact dynamical origin of these effects.

## Discussion

The recent deep and long-lived ozone holes have already resulted in extreme UV levels over Antarctica[45]. Beyond local UV effects, Antarctic ozone is intrinsically linked to the climate and dynamics of the Southern Hemisphere; changes in stratospheric ozone levels drive

circulation changes across the entire hemisphere, impacting the Hadley cell, the subtropical dry-zones, and the Southern Annular Mode[7–9]. We need to understand the changes taking place at the core of the ozone layer in order to determine how well these ongoing processes are captured in future model predictions.

By taking a holistic look at the yearly progression of the Antarctic ozone hole over the last two decades, we find that:

1. The addition of recent years to the Antarctic (60°S–90°S) total column ozone time series results in insignificant long-term change since the early 2000's, even where significant recovery has previously been reported. During this time, we find a delay in both the deep ozone hole onset date as well as the breakup date.

2. The early springtime upper stratosphere near the polar vortex edge shows signs of ozone recovery. However, from late September, the middle and lower stratospheric ozone poleward of 60°S declines at a rate of over − 0.03 ppmv/year from 2004 to 2022, with peak values in October exceeding − 0.10 ppmv/year.

3. At the core of the ozone hole (75°S–82°S, 5–50 hPa), the October partial column shows a total reduction of 26% from 2004 to 2022. This is equivalent to approximately 30 DU in the total column.

4. Atmospheric tracer observations show evidence of a vertical shift in mesospheric air descending into the core of the ozone hole. This is highly correlated with October stratospheric $NO_2$ and the ozone decline identified above.

The link we identify between the mid-spring ozone hole evolution and the timing/depth of mesospheric air arrival into the polar vortex sheds light on the potential drivers behind the recent large ozone holes. This mechanism is of dynamical origin and could be operating independently from the volcanic and wildfire effects implicated in the heightened ODS-induced ozone loss during the early springs of recent years[1,16,19,25]. More study is warranted to determine the precursors for the changes we identify in mesospheric descent and better understand the extent to which these effects will impact polar ozone in our changing climate.

## Methods
### Data and analysis for Fig. 1
Panel a: We use Total Ozone Mapping Spectrometer (TOMS) TCO data for the years 1979–2004 and the months of September, October, and November[46,47]. TOMS data is missing for Sept. and Oct. in 1993–1996 and for Nov. in 1993–1997, thus these months/years are excluded from the analysis. Gaps are shown in figures at the corresponding times. The latitude range 60°S–90°S is used to find monthly zonal mean TOMS TCO for each year present in the dataset[34].

The TOMS dataset is extended to 2022 using the Ozone Monitoring Instrument (OMI, on board the Aura-satellite) TOMS-Like Ozone L3 daily data from 2005 to 2022[48]. OMI data is missing for Sept. 28–29, 2008, thus these two days are excluded from the analysis. For each day of OMI measurements, the latitude range 60°S–90°S is used to find a zonal mean TCO[34]. The monthly mean TCO and standard deviation for each month are then calculated from the OMI daily mean TCO for each year present in the dataset. In the analysis, the years 2002 and 2019 are excluded, as they have been associated with sudden stratospheric warming events that dominate the formation and breakup of the ozone hole[1,18,49,50]. Linear regression with time as the sole predictor is performed from 2001 to 2022 to find the overall change in ozone for each month under investigation. We use 2001 as the starting point based on the theoretical pivot point in ozone depletion at the turn of the 21st century[34], however, changing the pivot year does not influence the overall results. The statistical significance is determined using the 95% confidence interval on the coefficient estimates. In addition, the $R^2$ value is reported.

Panel b: We again use the OMI TOMS-Like Ozone L3 daily data from 2005 to 2022[48] for best continuity with the TCO analysis in panel a. We first calculate a latitude-weighted area for all grid points from 50°S to 90 °S. Using a 130 DU ozone depth, as in Stone et al.[28], we find the total daily area (in $km^2$) of grid points which are less than the threshold. If there is missing data over the southernmost grid points, we check the furthest south grid point. If this grid point goes below the threshold, we assume that all missing grid points south of that point are also below the threshold. We then identify the first day when the area surpasses the 1 million $km^2$ area threshold and remains above for 3 days or more, again as in Stone et al.[28].

We additionally calculate the ozone hole breakup dates from 2005 to 2022 by finding the last day in the season when the < 130 DU ozone hole reaches an area above the 1 million $km^2$ threshold. The day after this is considered the breakup date.

### Data and analysis for Fig. 2
We use Microwave Limb Sounder (MLS, on-board the Aura satellite) Level 2 Ozone Mixing Ratio Version 5 data for 2004–2022 and for the months of September-November. MLS/Aura ozone data is known to have minimal drift over time when compared to ground-based detectors, so it is a reliable instrument to use when comparing ozone measurements over the 21st century[51]. The MLS ozone volume mixing ratio (VMR) data is limited to the recommended vertical range: 261 to 0.0215 hPa[52]. The data is interpolated to 1° × 1° latitude-longitude coverage. Outlier days excluded due to anomalous data are (reported as day of year) 2004: days 252 and 253; 2005: day 293; 2006: day 307; 2007: day 310; 2009: day 300; 2012: day 328; 2016: day 292; 2018: day 298; 2022: day 277.

Monthly means for the interpolated MLS/Aura ozone data are found for Sept., Oct., and Nov. for each year from 2004 to 2022. Then, a zonal average is calculated for every latitude across 45°S–82°S for each month/year and a linear regression fit to the resulting data is determined. As previously, data from 2019 is excluded from the fit[18]. The slope of each point is reported as change in parts per million volume ozone per year (ppmv/year). The 95% confidence interval, reported in the figure, is found using bootstrapping with 1000 iterations (results are consistent when 10,000 iterations are used).

### Analysis for Fig. 3
A daily zonal and latitudinal average from 75°S to 82°S is found in the daily MLS/Aura ozone profiles from August 21st through the end of November. A starting point of 75°S is used to reduce any interaction with the polar vortex[53]. Daily profiles are lined up sequentially from August 21st to November 30th to form a times series for each year. Data is stacked across years and a linear regression fit is found. Data from 2019 is excluded from the fit[18], as before. The slope of each point is reported as the change in parts per million volume ozone per year (ppmv/year), shown by contour colouring. The 95% confidence interval, reported in the figure, is found using bootstrapping as in the previous section.

### Analysis for Fig. 4
Four series are created from the October mean MLS/Aura data, each further isolated by a range of altitude and latitude levels: Pressure levels 5–50 hPa averaged from 60°S–82°S and from 75°S–82°S; Pressure levels 1–5 hPa averaged from 60°S–82°S and from 45°S to 60°S.

MLS ozone concentration values are then averaged zonally and across the latitude range specified to produce a single profile for each year, which is used to calculate a PCO. The resulting PCO values are used to find a linear regression fit for each time series. Statistical significance is reported using 95% confidence interval on the coefficient estimates from the regression fit, along with the $R^2$ value. An approximation of the net ozone loss through the range of years is calculated using the estimated slope of the linear fit.

## Data and analysis for Fig. 5

Panel a: We use Microwave Limb Sounder (MLS, on-board the Aura satellite) Level 2 Carbon Monoxide (CO) Mixing Ratio Version 5 data for 2004–2022 and for the months of August-November. The MLS CO volume mixing ratio (VMR) data is limited to the recommended vertical range: 215 to 0.00564 hPa[54]. Outlier days excluded due to anomalous data are (reported as day of year) 2004: days 252 and 253; 2009: day 299 and 300; 2018: day 298; 2022: day 277.

A daily zonal and latitudinal average from 75°S to 82°S is found in the daily MLS/Aura CO profiles from August 21st through the end of November. A starting point of 75°S is used to match the MLS/Aura ozone selection in Fig. 3. Daily profiles are lined up sequentially from August 21st to November 30th to form a times series for each year. Data is stacked across years and a linear regression fit is found. Data from 2019 is excluded from the fit, as before in Fig. 3[18]. The slope of each point is reported as the change in parts per billion volume CO per year (ppbv/year), shown by contour colouring. The 95% confidence interval, reported in the figure, is found using bootstrapping as in the previous sections.

Panel b: The MLS/Aura CO daily profiles are used for October 27th to October 31st (last 5 days of October) and are averaged zonally and latitudinally from 75°S to 82°S, as before in panel a. The altitude range is cropped to 5–50 hPa to match the coverage used for the PCO calculation for the 2nd series in Fig. 4. A Modified Akima Interpolation is performed to find the altitude (in kilometres) of the CO profile maximum for each day. The altitude of the CO maximum is averaged across the 5 days to find a single value for each year present in the dataset.

We use the Ozone Monitoring Instrument (OMI, on board the Aura-satellite) NO2 Stratospheric MINDS Daily L2 daily October data from 2005 to 2022[55]. For each day of OMI measurements, the latitude range 75°S–90°S is used to find a zonal mean stratospheric $NO_2$ concentration in mole/cm$^2$ × 10$^{15}$. October mean stratospheric $NO_2$ is then calculated from the OMI daily mean $NO_2$ for each year present in the dataset.

The October mean stratospheric $NO_2$ versus the altitude of late-October CO maximum is used to find a linear regression fit across 2005–2022 (2004 is excluded from the CO data due to the starting point of 2005 for the OMI $NO_2$ data). Statistical significance is reported using 95% confidence interval on the coefficient estimates from the regression fit, along with the $R^2$ value.

Panel c: The MLS/Aura altitude of late-October CO maximum data is used again, as calculated in panel b.

We select the 2nd PCO series produced in Fig. 4, using the pressure level range of 5–50 hPa and averaged from 60°S to 82°S, across 2004–2022.

The October PCO (from 5 to 50 hPa and 60°S–82°S) versus the altitude of late-October CO maximum is used to find a linear regression fit across 2004–2022. Statistical significance is reported using 95% confidence interval on the coefficient estimates from the regression fit, along with the $R^2$ value.

## Data availability

All data used in this study is freely available from the following sources: The TOMS Earth Probe Total Column Ozone data are available at https://disc.gsfc.nasa.gov/datasets/TOMSN7L3mtoz_008/summary?keywords=TOMS and https://disc.gsfc.nasa.gov/datasets/TOMSEPL3mtoz_008/summary?keywords=TOMS[46,47]. The OMI/Aura TOMS-Like Ozone L3 data are available at https://disc.gsfc.nasa.gov/datasets/OMTO3e_003/summary?keywords=OMI. The MLS/Aura Level 2 Ozone Mixing Ratio Version 5 data are available at https://disc.gsfc.nasa.gov/datasets/ML2O3_005/summary?keywords=mls. The MLS/Aura Level 2 CO Mixing Ratio Version 5 data are available at https://disc.gsfc.nasa.gov/datasets/ML2CO_005/summary?keywords=mls. The OMI/Aura NO2 Stratospheric MINDS Daily L2 data are available at https://disc.gsfc.nasa.gov/datasets/OMI_MINDS_NO2G_1.1/summary?keywords=omi[55].

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

## Acknowledgements

H.E.K. and A.S. were partially funded by the New Zealand Ministry of Business, Innovation & Employment Endeavour fund Smart Ideas project PROP-76111-ENDSI-UOO (Contract UOOX2106).

## Author contributions

H.E.K. and A.S. planned the study. H.E.K. analysed the data and prepared the figures, with suggestions from A.S. and C.J.R. H.E.K. and A.S. wrote the manuscript with comments from C.J.R.

## Competing interests

The authors declare no competing interests.
