## [Peer Review File · Nature Communications]

Potential drivers of the recent large Antarctic ozone holesREVIEWER COMMENTS

Reviewer #1 (Remarks to the Author):

Review of “Polar stratospheric ozone continues on a decline: A complex story of the changing Antarctic ozone hole” by Kessenich et al. (2023) submitted to Nature communications.

Based on MLS satellite, Kessenich et al. (2023) analyzed the observed ozone trends in the middle and upper stratosphere in Southern Hemisphere (SH). The major findings are 1) with the recent two years' data, the Antarctic stratospheric total column ozone (TCO) trends are all negative and insignificant; 2) the ozone trend is negative in the Antarctic middle stratosphere, and the ozone PCO decrease by 26% (or ~30 DU) from 2004 to 2020.

By including the 2020-2022's data, the Antarctic ozone trend from 2004 to 2020 become negative. It is not surprising as we know at least wildfire and volcanic eruptions amid 2019-2022 contributed to the observed change. From the title of “A complex story of the changing Antarctic ozone hole”, I expected the authors analyzing and quantifying the causes of the observed ozone change in the Antarctic stratosphere. Unfortunately, I failed to find sufficient discussions either from observation or the modeling point of view. I also failed to find analysis how much the ozone recovery year can be affected. What caused the negative trend in the middle stratosphere since 2004? Is it dynamics-driven or chemistry-driven?

I do think the Antarctic ozone trend is an important issue, and I don't see any major technique problem of the analysis in the manuscript. However, I worry the analysis and conclusions present in the manuscript are superficial.

Minor comments:

1. It is helpful to add the Line number in the manuscript.
2. “Our findings suggest that changes in the SH atmosphere are contributing to a persistent Antarctic ozone hole. ” This is an ambiguous statement. What are the “changes”?

Reviewer #2 (Remarks to the Author):

This study uses vertically-resolved observations of ozone to focus on changes in the Antarctic ozone hole over the period 2001 to 2022. The authors claim that their results support their conclusions that 1) Antarctic ozone is not recovering significantly and 2) the region and month of strongest ozone decline occurs over the polar middle and lower stratosphere during October. As a result of 1) and 2), the authors

conclude that dynamical factors are resulting in a persistent SH ozone hole that is not showing any statistically significant signs of recovery. Given that this conclusion is quite bold (as it questions the efficacy of the Montreal Protocol) I would like to see how this result bears up when other metrics of the ozone hole are considered (major comment 1 below). I strongly recommend this as I suspect that a more exhaustive analysis will reveal subtleties that may challenge the authors' main conclusions and, if anything, may lead to the opposite conclusion. I also have concerns about the mechanism for continued ozone decline that is (hastily) proposed in the conclusions as no evidence for this mechanism (from either modeling or observations) is presented. To this end, I suggest major revisions before I can begin consider recommending publication.

Major Comment 1: Ozone hole onset date: One of the references cited in the study, Kane et al. (2021), points out that, despite the recent large SH ozone holes of 2015, 2018, 2020 and 2021 (also noted in this study), there has been a delay in ozone hole onset since 2020 (see their Figure 2). The implication, therefore, is that "even though 2015, 2018, 2020, and 2021 had large ozone holes, there is a clear later onset of ozone depletion in these years compared to 1999–2008 (the peak in stratospheric EESC loading) expressing that ozone recovery is still proceeding following the decline in polar EESC since ~2000 (WMO, World Meteorological Organization, 2018)." As a result, the authors interpret this later start date for the rapid onset of ozone depletion as "a robust sign of ozone recovery post-2000."

Request: Can the authors examine ozone hole onset date and corroborate the findings from this study? If so, I have to admit that I agree with the Kane et al. (2021) assessment that ozone loss must be considered in the context of the formation date of the ozone hole and that, furthermore, a delayed ozone hole onset could be interpreted as a sign of ozone recovery. This should be discussed more in the present study and discussed with respect to how this may change the authors' major conclusion.

Stone, K.A., Solomon, S., Kinnison, D.E., Mills, M.J.: On Recent Large Antarctic Ozone Holes and Ozone Recovery Metrics. *Geophysical Research Letters* 48(22), 2021–095232 (2021). <https://doi.org/10.1029/2021GL095232>

Major Comment 2: The proposed mechanisms for October ozone decreases seems rather hastily thrown in the conclusions and unsupported. In particular, the authors suggest that the ozone reductions are "linked to an air mass descending from late August from around 1-2 hPa." There is no observational or modeling evidence to support this claim. My request, therefore, is that the authors either 1) run modeling experiments integrating an age-of-air tracer which would capture the transport phenomenon that they propose is occurring or 2) provide evidence from the observations using observationally-based transport metrics. If they cannot provide either 1) or 2), then the manuscript loses a significant part of its novelty and it's not clear that the results presented substantiate publication in a high-impact journal like *Nature*.

RESPONSE TO REVIEWER COMMENTS

We would like to thank both reviewers for providing their valuable comments on our manuscript. After reviewing their recommendations in detail, we have incorporated a number of changes into our new draft. The reviewers' input has helped to strengthen the quality of our manuscript, particularly in terms of further evidence on drivers for the October ozone changes. Our detailed responses to all reviewer comments are included below. We note that some formatting changes were made in revision to meet the journal formatting guidelines.

Reviewer #1 (Remarks to the Author):

Overall Comment

Based on MLS satellite, Kessenich et al. (2023) analyzed the observed ozone trends in the middle and upper stratosphere in Southern Hemisphere (SH). The major findings are 1) with the recent two years' data, the Antarctic stratospheric total column ozone (TCO) trends are all negative and insignificant; 2) the ozone trend is negative in the Antarctic middle stratosphere, and the ozone PCO decrease by 26% (or ~30 DU) from 2004 to 2020.

By including the 2020-2022's data, the Antarctic ozone trend from 2004 to 2020 become negative. It is not surprising as we know at least wildfire and volcanic eruptions amid 2019-2022 contributed to the observed change. From the title of "A complex story of the changing Antarctic ozone hole", I expected the authors analyzing and quantifying the causes of the observed ozone change in the Antarctic stratosphere.

Unfortunately, I failed to find sufficient discussions either from observation or the modeling point of view. I also failed to find analysis how much the ozone recovery year can be affected. What caused the negative trend in the middle stratosphere since 2004? Is it dynamics-driven or chemistry-driven?

I do think the Antarctic ozone trend is an important issue, and I don't see any major technique problem of the analysis in the manuscript. However, I worry the analysis and conclusions present in the manuscript are superficial.

Response

We have revised the manuscript by adding more satellite data analysis to particularly address the differences between chemistry-driven and dynamics-driven ozone loss. For the latter, we include more observational evidence from the Aura and OMI instruments as well as discussion of dynamical drivers that influence the ozone loss patterns reported in the manuscript. More details on these are included in the responses to specific questions below.

In terms of the ozone recovery year, our aim was not to reassess the current prediction. Rather we are focused on the complex interplay of chemistry and dynamics of the Antarctic atmosphere, which itself is often less of a focus on large assessments looking at overall global ozone levels, or annual scale recovery. Our aim is to specifically focus on the spatiotemporal variability of Antarctic stratospheric ozone, as this is not captured for total ozone column assessment or many of the other ozone recovery metrics commonly used.

We now address the reviewer's primary request to present evidence for the cause of the negative ozone trend in the middle stratosphere with the addition of a new section titled "Drivers of the changes in ozone" and Figure 5 to our manuscript. Further related details are also included in our responses to specific points raised by Reviewer #2, included below.

Minor Comment One

It is helpful to add the Line number in the manuscript.

Response

Line numbers have been added as requested.

Minor Comment Two

“Our findings suggest that changes in the SH atmosphere are contributing to a persistent Antarctic ozone hole.” This is an ambiguous statement. **What are the “changes”?**

Response

We have rewritten the last conclusion paragraph to be more specific regarding the dynamic changes in mesospheric descent we see in the polar middle stratosphere. This arises from the new analysis added in the new Figure 5 and the section titled “Drivers of the changes in ozone” of our manuscript. With more specific findings that are now discussed, this previous ambiguity regarding the “changes” has been replaced by more details.

For example, we have added new text for the last paragraph of the conclusion which reads:

“By taking a holistic look at the yearly progression of the Antarctic ozone hole over the last two decades, our results shed new light on the deep and long-lived ozone holes of recent years. *We identify links between the mid-spring ozone hole evolution and the timing/depth of mesospheric air arrival into the polar vortex.* This mechanism is of dynamical origin and could be operating independently from the volcanic and wildfire effects implicated for the heightened ODS-induced ozone loss during the early springs of recent years [1, 6, 20, 23]. More study is warranted to determine the precursors for the changes we identify in mesospheric descent and better understand the extent to which these effects will impact polar ozone in our changing climate.”

Reviewer #2 (Remarks to the Author):

Overall Comment

This study uses vertically-resolved observations of ozone to focus on changes in the Antarctic ozone hole over the period 2001 to 2022. The authors claim that their results support their conclusions that 1) Antarctic ozone is not recovering significantly and 2) the region and month of strongest ozone decline occurs over the polar middle and lower stratosphere during October. As a result of 1) and 2), the authors conclude that dynamical factors are resulting in a persistent SH ozone hole that is not showing any statistically significant signs of recovery. Given that this conclusion is quite bold (as it questions the efficacy of the Montreal Protocol) I would like to see how this result bears up when other metrics of the ozone hole are considered (major comment 1 below). I strongly recommend this as I suspect that a more exhaustive analysis will reveal subtleties that may challenge the authors' main conclusions and, if anything, may lead to the opposite conclusion. I also have concerns about the mechanism for continued ozone decline that is (hastily) proposed in the conclusions as no evidence for this mechanism (from either modeling or observations) is presented. To this end, I suggest major revisions before I can begin consider recommending publication.

Response

Our intention is not to question the efficacy of the Montreal Protocol, nor do we believe our results refute that. Indeed, we included analysis of the total column ozone that continues the long timeseries that has been published many times previously. As we show in our Figure 1, September total column ozone continues to indicate recovery. The effectiveness of the ban on CFCs and the impact on the early spring total column ozone is a positive development, and, as we also show, this recovery is mainly focused on the upper stratosphere and the poleward side of the polar vortex (Figure 2). We do highlight that this progress is yet to reach the core of the stratospheric ozone layer.

Our original aim was to focus on the changes in Antarctic ozone regardless of whether these were caused by CFC related ozone depletion or other mechanisms. The recent large ozone holes have a significant impact on both UV levels and climate and dynamics of the Southern Hemisphere. We need to understand these extended periods of large ozone holes to be able to understand how well these processes are captured in regional variability as well as future model predictions for our part of the world (the authors reside in New Zealand). We very much agree that both reviewers raise a good point, and we acknowledge that identifying the likely cause behind the changes we find is an important next step and should be addressed in this manuscript. Thus, we have added more data and analysis to support our previous results – we detail the new additions in the following responses. This additional analysis has helped sharpen the conclusions of the work and we have also rewritten the text in the manuscript to try and make sure the manuscript is clear in that we do not refute the progress from the Montreal Protocol.

Major Comment One

Ozone hole onset date: One of the references cited in the study, Kane et al. (2021), points out that, despite the recent large SH ozone holes of 2015, 2018, 2020 and 2021 (also noted in this study), there has been a delay in ozone hole onset since 2020 (see their Figure 2). The implication, therefore, is that “even though 2015, 2018, 2020, and 2021 had large ozone holes, there is a clear later onset of ozone depletion in these years compared to 1999–2008 (the peak in stratospheric EESC loading) expressing that ozone recovery is still proceeding following the decline in polar EESC since ~2000 (WMO, World Meteorological Organization, 2018).” As a result, the authors interpret this later start date for the rapid onset of ozone depletion as “a robust sign of ozone recovery post-2000.”

Request: **Can the authors examine ozone hole onset date and corroborate the findings from this study?** If so, I have to admit that I agree with the Kane et al. (2021) assessment that ozone loss must be considered in the context of the formation date of the ozone hole and that, furthermore, a delayed ozone hole onset could be

interpreted as a sign of ozone recovery. This should be discussed more in the present study and discussed with respect to how this may change the authors' major conclusions.

Stone, K.A., Solomon, S., Kinnison, D.E., Mills, M.J.: On Recent Large Antarctic Ozone Holes and Ozone Recovery Metrics. *Geophysical Research Letters* 48(22), 2021–095232 (2021). <https://doi.org/10.1029/2021GL095232>

Response

The ozone hole onset date is a beneficial metric to consider, and we agree with the reviewer that it would make a good addition to our analysis, thus we have added this information to the revised Figure 1. Stone et al. (2021) find that even large ozone holes of recent years (2015 and 2020) show a shift towards later onset dates, and hence support the continued recovery from CFC-induced ozone loss. We find that this metric goes hand-in-hand with the TCO analysis in Figure 1 of our manuscript.

We calculated the ozone hole onset date from 2005-2022 using a similar method as Stone et al. (2021) for the TCO thresholds of 130 DU, 175 DU, and 220 DU. As we are focused on the very deep ozone holes of recent years, we included the results for the 130 DU threshold trend in Figure 1b in the manuscript. The results for the 175 DU and 220 DU onset dates are included here. All robustly show a delay in the onset date, consistent with Stone et al. (2021).

We also calculated the deep ozone hole breakup dates for each of the thresholds (included in the figures below). This date was identified as the day after the last time the ozone hole surpasses the threshold during the season. While the breakup dates show more variability over time, the recent years in particular have presented a clear delay though all the thresholds we considered.

Furthermore, assessing the onset date and breakup date together on one figure emphasizes that both metrics show a similar degree of delay, or shift, during 2005-2022. Recent years have also seen some of the largest “windows” of deep ozone holes throughout the period analyzed.

By adding this ozone hole onset and breakup date assessment to our new Figure 1b, this helps differentiate the signs of recovery we find in September from the declines in ozone during October and November.

We have revised Figure 1 to the version shown below and now write in the main manuscript text:

“The trend in the onset date (green line in Fig. 1b) agrees with previous findings of a delayed ozone hole opening date [1, 2, 16, 29, 34]. This delay is likely an indication of early-spring recovery due to EESC reduction [1, 16]. Our results for the breakup date trend (magenta line in Fig.1b) show a similar delay in the breakup of the ozone hole, albeit much noisier and insignificant. Given that the deep ozone hole breakup typically occurs during October/November, a later breakup date is in agreement with the declines in October/November TCO in Figure 1a. These results suggest that there may be new patterns emerging in the mid-spring evolution of the ozone hole, thus delaying the breakup date.”

New manuscript Figure 1:

Fig. 1 (a) Zonal mean total column ozone (TCO) in Dobson units (DU) across latitudes 60°S to 90°S shown for the months of Sept., Oct., and Nov. from TOMS/OMI data for years 1979–2022 (2002 and 2019 excluded). The lines present a linear fit from 2001–2022, with the fit equation shown in the legend. The linear fit uncertainty is quoted at the 95% confidence interval and $T = \text{Years since 2001}$. (b) Deep ozone hole onset and breakup dates (130 DU, 1 million km²) from OMI data for years 2005–2022. Missing years do not surpass the TCO/area threshold during the season. The lines present a linear fit from 2005–2022, with the fit equation shown in the legend. The linear fit uncertainty is quoted at the 95% confidence interval and $T = \text{Years since 2005}$.

175/220 DU onset date figures (not included in manuscript):

Major Comment Two

The proposed mechanisms for October ozone decreases seems rather hastily thrown in the conclusions and unsupported. In particular, the authors suggest that the ozone reductions are **“linked to an air mass descending from late August from around 1-2 hPa.”** **There is no observational or modeling evidence to support this claim.** My request, therefore, is that the authors either **1) run modeling experiments integrating an age-of-air tracer which would capture the transport phenomenon that they propose is occurring or 2) provide evidence from the observations using observationally-based transport metrics.** If they cannot provide either 1) or 2), then the manuscript loses a significant part of its novelty and it’s not clear that the results presented substantiate publication in a high-impact journal like Nature.

Response

This is a very valuable point, and we would like to thank the reviewer for raising this. We have now addressed this by adding in satellite observations of an atmospheric tracer to assess the potential dynamical drivers. Overall, the new data has sharpened the picture of the role of dynamics, and changes in vertical descent that we propose to provide an explanation to the recent trends in October ozone levels. We discuss this and the new analysis in more detail below.

As we wrote above as a response to the overall comment, it was our original intention to focus on the changes in Antarctic ozone regardless of whether these were caused by chlorine/bromine ozone depletion or other mechanisms. The issue of CFC-related ozone recovery is often the only focus of published studies, while the large ozone holes of recent years are still impacting UV and circulation in the Southern Hemisphere. The reviewers raise a good point, and we do agree that identifying the likely cause behind the changes we find is an important next step and should be addressed in the manuscript. As a response we have added a new figure, Figure 5, as well as a section titled, “Drivers of the changes in ozone” in the revised manuscript. The new figure is also provided below for reference.

Panel a in Figure 5 provides observational evidence of changes to descending air mass from higher altitudes. We choose to use observational evidence, in the form of the atmospheric tracer carbon monoxide (CO) from MLS/Aura, as this links directly to the MLS data we already used in the manuscript.

The MLS CO shows air descending from the mesosphere into the spring polar vortex. Our new figure identifies changes in the timing and altitude of the descending CO, with signs of a shift towards later/shallower mesospheric descent in recent years. The regions where this shift in CO is statistically significant line up with regions of significant negative changes in ozone in the October middle stratosphere that we identified earlier in Figures 3 and 4. This would indicate that there are changes aloft which are of dynamical origin occurring during October.

We also include a new correlation result in Figure 5, panel b, which compares the carbon monoxide altitude at the end of October to OMI/Aura observations of the stratospheric NO₂ levels during October. Previous work has shown that, in addition to transport from lower latitudes, within the polar vortex NO₂ (or NO_x) has an important source in the mesosphere, which is known to descend into the polar stratosphere during the springtime. Once in the stratosphere, it can both destroy ozone and impact the timing of chlorine and bromine reservoir formation. We show that the two are highly correlated, indicating a relationship between the timing and depth of the descending air within the polar vortex and the amount of NO₂ (and likely the NO_x family of chemicals) arriving in the October stratosphere.

Finally, we include another new correlation result in Figure 5, panel c, comparing the carbon monoxide altitude at the end of October to the high latitude middle stratosphere partial column ozone from Figure 4 in our manuscript. This PCO window covers the region of declining ozone we identified in Figure 3 of our original manuscript. The two are highly correlated, indicating a relationship between the timing and depth of the descending air within the polar vortex and ozone concentrations at the core of the ozone hole.

Together these new results provide evidence that the October ozone changes are linked to changes in vertical transport of airmasses. These dynamical changes are taking place alongside of volcanic eruptions and wildfires and can now help explain the extend of ozone loss observed in the recent years.

We propose that this mechanism should be investigated further in a follow-on study due to its complexity. We hope that the links we identify in Figure 5 and the section titled, “Drivers of the changes in ozone” are sufficient to address the possible drivers of recent declines in ozone we present.

We do note here that careful considerations would need to go into any model studies looking into this as many models have been known to underestimate the descent of mesospheric air inside the polar vortex, and do not routinely capture the high altitude production of NO_x.

New manuscript Figure 5:

REVIEWERS' COMMENTS

Reviewer #1 (Remarks to the Author):

Kessenich et al. has done an excellent job in addressing my and reviewer2's comment. A new figure (Fig.5) and new section has been added in the manuscript discussing the potential causes of the ozone decline. I found the fig5 very interesting and seems the dynamical variability of the descending of the mesospheric air is of quite important role in polar ozone. Although the effect on the PCO/TCO has still not been explicitly quantified, and I agree with the authors that probably it is not that easy to model at this moment. I am happy to recommend this manuscript for publication.

btw, Figure5 legend: "and T = years since 2005."

I don't see T in Fig5.

Reviewer #2 (Remarks to the Author):

The reviewers have done a good job incorporating and responding to my major comments. In particular, I appreciate the 1) new analysis of breakup and onset dates and 2) CO evolution which, respectively, better place their results in the context of previous studies and suggest a possible dynamical origin to the observed O3 trends. Both contributions strengthen the manuscript.

One lingering issue, however, regards the greater impacts of these findings, specifically on whether these results qualify for a higher-impact journal. In particular, it is clear now that the authors corroborate the findings from the Kane et al. (2021) study showing a later ozone onset date, which some readers may choose to interpret as a sign of ozone recovery (not depletion). The question then becomes why do we care about such a localized (~10-50 hPa) anomaly? I believe the authors do have an answer to this, as they explain in their rebuttal that "The recent large ozone holes have a significant impact on both UV levels and climate and dynamics of the Southern Hemisphere. We need to understand these

extended periods of large ozone holes to be able to understand how well these processes are captured in regional variability as well as future model predictions for our part of the world (the authors reside in New Zealand)." My recommendation, therefore, is that the authors include a new paragraph in the conclusions stating something along these lines so that the readers clearly understand why the reported results matter.

Once the additional material addressing the impact of the results is added, I am inclined to recommend publication (pending, of course, that the comments from the other reviewer are addressed).

RESPONSE TO REVIEWER COMMENTS

We would like to thank both reviewers once again for providing their valuable comments on the revised version of our manuscript. Our responses to all reviewer comments are included below.

Reviewer #1 (Remarks to the Author):

Overall Comment

Kessenich et al. has done an excellent job in addressing my and reviewer2's comment. A new figure (Fig.5) and new section has been added in the manuscript discussing the potential causes of the ozone decline. I found the fig5 very interesting and seems the dynamical variability of the descending of the mesospheric air is of quite important role in polar ozone. Although the effect on the PCO/TCO has still not been explicitly quantified, and I agree with the authors that probably it is not that easy to model at this moment. I am happy to recommend this manuscript for publication.

Minor Comment One

btw, Figure5 legend: "and T = years since 2005."
I don't see T in Fig5.

Response

We appreciate the reviewer bringing this to our attention, as the "T = years since 2005" text in Fig. 5 was included in error. We have edited the caption to remove this text.

Reviewer #2 (Remarks to the Author):

Overall Comment

The reviewers have done a good job incorporating and responding to my major comments. In particular, I appreciate the 1) new analysis of breakup and onset dates and 2) CO evolution which, respectively, better place their results in the context of previous studies and suggest a possible dynamical origin to the observed O3 trends. Both contributions strengthen the manuscript.

Major Comment One

One lingering issue, however, regards the greater impacts of these findings, specifically on whether these results qualify for a higher-impact journal. In particular, it is clear now that the authors corroborate the findings from the Kane et al. (2021) study showing a later ozone onset date, which some readers may choose to interpret as a sign of ozone recovery (not depletion). The question then becomes why do we care about such a localized (~10-50 hPa) anomaly? I believe the authors do have an answer to this, as they explain in their rebuttal that "The recent large ozone holes have a significant impact on both UV levels and climate and dynamics of the Southern Hemisphere. We need to understand these extended periods of large ozone holes to be able to understand how well these processes are captured in regional variability as well as future model predictions for our part of the world (the authors reside in New Zealand)." **My recommendation, therefore, is that the authors include a new paragraph in the conclusions stating something along these lines so that the readers clearly understand why the reported results matter.**

Once the additional material addressing the impact of the results is added, I am inclined to recommend publication (pending, of course, that the comments from the other reviewer are addressed).

Response

The reviewer raises a great point, as the issue of the impact of our findings was an important topic to address when replying to the reviewer's comment. Following that logic, we highly agree that addressing this question within the body of our manuscript as well is of similar importance. A new paragraph addressing the impacts of our results has been added within the "Discussion" section (previously "Conclusions", the change was due to editorial requirements). See the text below for the new Discussion section, with the added text in bold:

"The recent deep and long-lived ozone holes have already resulted in extreme UV levels over Antarctica [45]. Beyond local UV effects, Antarctic ozone is intrinsically linked to the climate and dynamics of the Southern Hemisphere; changes in stratospheric ozone levels drive circulation changes across the entire hemisphere, impacting the Hadley cell, the subtropical dry-zones, and the Southern Annular Mode [7–9]. We need to understand the changes taking place at the core of the ozone layer in order to determine how well these now ongoing processes are captured in future model predictions.

By taking a holistic look at the yearly progression of the Antarctic ozone hole over the last two decades, we find that:

1. The addition of recent years to the Antarctic (60°S–90°S) total column ozone time series results in insignificant long term change since the early 2000's, even where significant recovery has previously been reported. During this time, we find a delay in both the deep ozone hole onset date as well as the breakup date.
2. The early springtime upper stratosphere near the polar vortex edge shows signs of ozone recovery. However, from late September, the middle and lower stratospheric ozone poleward of 60°S declines at a rate of over -0.03 ppmv/year from 2004 to 2022, with peak values in October exceeding -0.10 ppmv/year.
3. At the core of the ozone hole (75°S–82°S, 5–50 hPa), the October partial column shows a total reduction of 26% from 2004 to 2022. This is equivalent to approximately 30 DU in the total column.
4. Atmospheric tracer observations show evidence of a vertical shift in mesospheric air descending into the core of the ozone hole. This is highly correlated with October stratospheric NO₂ and the ozone decline identified above.

The link we identify between the mid-spring ozone hole evolution and the timing/depth of mesospheric air arrival into the polar vortex sheds light on the potential drivers behind the recent large ozone holes. This mechanism is of dynamical origin and could be operating independently from the volcanic and wildfire effects implicated for the heightened ODS-induced ozone loss during the early springs of recent years [1, 16, 19, 25]. More study is warranted to determine the precursors for the changes we identify in mesospheric descent and better understand the extent to which these effects will impact polar ozone in our changing climate."